# Design, Synthesis and Functional Analysis of Cyclic Opioid Peptides with Dmt-Tic Pharmacophore

**DOI:** 10.3390/molecules25184260

**Published:** 2020-09-17

**Authors:** Arijit Sarkar, Anna Adamska-Bartlomiejczyk, Justyna Piekielna-Ciesielska, Karol Wtorek, Alicja Kluczyk, Attila Borics, Anna Janecka

**Affiliations:** 1Laboratory of Chemical Biology, Institute of Biochemistry, Biological Research Centre, Szeged, 62. Temesvári krt., H-6726 Szeged, Hungary; sarkar.arajit@brc.hu (A.S.); borics.attila@brc.hu (A.B.); 2Theoretical Medicine Doctoral School, Faculty of Medicine, University of Szeged, 97. Tisza L. krt., H-6722 Szeged, Hungary; 3Department of Biomolecular Chemistry, Faculty of Medicine, Medical University of Lodz, Mazowiecka 6/8, 92-215 Lodz, Poland; anna.adamska-bartlomiejczyk@umed.lodz.pl (A.A.-B.); justyna.piekielna-ciesielska@umed.lodz.pl (J.P.-C.); karol.wtorek@umed.lodz.pl (K.W.); 4Faculty of Chemistry, University of Wroclaw, F. Joliot-Curie 14, 50-383 Wroclaw, Poland; alicja.kluczyk@chem.uni.wroc.pl

**Keywords:** peptide synthesis, functional assay, molecular modeling, opioid receptors, δ opioid receptor antagonists

## Abstract

The opioid receptors are members of the G-protein-coupled receptor (GPCR) family and are known to modulate a variety of biological functions, including pain perception. Despite considerable advances, the mechanisms by which opioid agonists and antagonists interact with their receptors and exert their effect are still not completely understood. In this report, six new hybrids of the Dmt-Tic pharmacophore and cyclic peptides, which were shown before to have a high affinity for the µ-opioid receptor (MOR) were synthesized and characterized pharmacologically in calcium mobilization functional assays. All obtained ligands turned out to be selective antagonists of the δ-opioid receptor (DOR) and did not activate or block the MOR. The three-dimensional structural determinants responsible for the DOR antagonist properties of these analogs were further investigated by docking studies. The results indicate that these compounds attach to the DOR in a slightly different orientation with respect to the Dmt-Tic pharmacophore than Dmt-TicΨ[CH_2_-NH]Phe-Phe-NH_2_ (DIPP-NH_2_[Ψ]), a prototypical DOR antagonist peptide. Key pharmacophoric contacts between the DOR and the ligands were maintained through an analogous spatial arrangement of pharmacophores, which could provide an explanation for the predicted high-affinity binding and the experimentally observed functional properties of the novel synthetic ligands.

## 1. Introduction

Opioid receptors (µ, δ, κ or MOR, DOR, KOR) belong to the G-protein-coupled receptor family [1]. They are found mostly in the central and peripheral nervous systems and in the gastrointestinal tract [2]. Among the three main types of opioid receptors, MOR is the one mostly associated with analgesia but also with the occurrence of the well-known adverse effects, including the inhibition of respiratory and gastrointestinal functions and the development of physical dependence and tolerance [3,4]. DOR is known to play a role in opioid-mediated side effects, including analgesic tolerance [5].

MOR and DOR are co-localized in pain inhibitory pathways [6] and the functional and physical cross-interactions between these two receptors have been well documented [7]. The combined administration of morphine with DOR antagonists was shown to cause attenuation of tolerance [8]. In DOR knock-out mice, morphine retained its MOR-mediated analgesia without the development of tolerance upon chronic administration [9,10,11]. These observations led to the design of multitarget ligands that would stimulate MOR in conjunction with antagonism at DOR.

Joining MOR-agonist and DOR-antagonist properties in one molecule can be achieved by the design of either bifunctional or bivalent ligands [12]. Bifunctional ligands are non-selective and possess highly integrated pharmacophores that can simultaneously activate two targets. Bivalent ligands contain two distinct pharmacophores connected either directly or via a linker, each of them able to interact with a different receptor [13,14,15,16].

A lot of ligands, either of a peptide or non-peptide structure, with an MOR agonist/DOR antagonist profile have been synthesized in the effort to obtain strong analgesics with little or no tolerance development [17,18]. The first bifunctional ligands with such properties were discovered among the analogs of endomorphin-2 (EM-2, Tyr-Pro-Phe-Phe-NH_2_) [19]. The replacement of Tyr by Dmt (2’,6’-dimethyl-l-tyrosine), Pro by Tic (Tic = 1,2,3,4-tetrahydroisoquinoline-3-carboxylic acid) and reduction in the Tic-Phe peptide bond led to the analog Dmt-TicΨ[CH_2_-NH]Phe-Phe-NH_2_ (DIPP-NH_2_[Ψ]), which was shown to reduce acute tolerance in rats when compared to morphine administration [20]. It was then established that the Dmt-Tic pharmacophore represents the shortest sequence that can still selectively interact with DOR as a potent antagonist [21]. Dmt-Tic was embedded into the sequence of various MOR-selective ligands, producing compounds with interesting biological activities. EM-2 analog, Dmt-Tic-(2*R*,3*S*)AHPBA-Phe-NH_2_, incorporating α-hydroxy-β-phenylalanine (AHPBA) showed an MOR agonist/DOR partial agonist profile and in the in vivo studies in mice produced comparable with morphine antinociception but did not demonstrate acute antinociceptive tolerance [22]. The distance between the Dmt-Tic pharmacophore and the third aromatic residue in the peptide sequence was found to be responsible for converting potent Dmt-Tic-based DOR-antagonists into non-selective ligands with an MOR agonist/DOR antagonist profile. Dmt-Tic-Gly-NH-Bzl (UFP-505) exhibited very high MOR-agonism and potent DOR-antagonism [23]. EM-2 extended at the C-terminus with Dmt-Tic via the ethylenediamine linker gave a bivalent analog Tyr-Pro-Phe-Phe-NH-CH_2_-CH_2_-NH←Tic←Dmt, in which both moieties retained their inherent opioid receptor preference and biological activities [24]. However, interesting in vitro activities of Dmt-Tic-containing analogs did not result in their development as novel analgesics, due to such shortcomings as poor water solubility, inability to cross the blood–brain barrier, unbalanced MOR/DOR affinity.

Structural requirements of binding to the MOR and DOR have been studied extensively in the past decades [25,26,27,28]. The recently solved crystallographic structures of these receptors in the presence of agonists or antagonists of various molecular scaffolds presented a great leap towards the understanding of the basic principles of opioid activity [29,30,31,32,33,34]. A common feature of receptor–ligand interactions is the presence of an anchoring salt bridge between Asp^3.32^ of the receptors and the protonated *N*-terminal primary amine of opioid peptides or the basic nitrogen atom of opioid alkaloids. In fact, the positioning of this basic nitrogen in the binding pocket and the arrangement of polar species around Asp^3.32^ was suggested to distinguish DOR agonists from antagonists. This hypothesis is, however, not entirely applicable in case of the MOR. An interaction between the phenolic OH group of the tyramine moiety of opioid alkaloids or the *N*-terminal Tyr side chain of opioid peptides and an extended network of polar amino acid side chains and structural water molecules was proposed to be the key feature of the active functional state of this receptor. Several other relevant pharmacophoric contacts were identified, such as, for example, a direct hydrogen bond observed frequently between the *N*-terminal primary amine of opioid peptides and Tyr^7.43^ of the receptors, but the majority of those contacts are specific to the given opioid receptor and ligand type. Therefore, structural determinants of opioid receptor activation, concerning both the receptors and their ligands, remain elusive.

Identifying structural features required for binding to MOR and DOR but activating only one receptor may be of a great help in the design of future drug candidates. In this study, we report the synthesis and pharmacological assessment of several bivalent ligands combining cyclic structures that were previously shown to possess high MOR affinity [35,36] with the Dmt-Tic fragment, known to confer DOR-antagonist properties.

## 2. Results

### 2.1. Chemistry

Six novel chimeric peptides combining our previously reported cyclic scaffolds with Dmt-Tic at the N-terminus or on both termini were designed (Table 1). The cyclic fragments were chosen on the basis of their high affinity for MOR (in the nM range) [36,37,38].

Peptides were successfully synthesized utilizing the standard solid-phase procedure using Fmoc/tBu chemistry with the hyper-acid labile Mtt/*O*-2-PhiPr groups for the selective protection of amine/carboxyl side-chain groups engaged in the formation of the cyclic fragment. Introduction of Tic into the peptide chain was more challenging, probably because of the more rigid structure of this amino acid. Successful incorporation of this residue was achieved through the application of a triazine-based coupling reagent, 4-(4,6-dimethoxy-1,3,5-triazin-2-yl)-4-methylmorpholinium toluene-4-sulfonate (DMT/NMM/TsO^−^) [39] instead of the generally used 2-(1H-benzotriazol-1-yl)-1,1,3,3-tetramethyluronium tetrafluoroborate (TBTU).

On-resin cyclization yielded mixtures of cyclic monomers and dimers. In case of analogs **1**–**3**, containing 17-membered rings, the monomer/dimer ratios were around 3/1. For analogs **4**–**6**, with smaller, 14-membered rings, these ratios were much lower. In case of analog **5**, the cyclic dimer was the main product. The yields of analogs **1**–**6** in the crude mixtures are given in Table 1.

All compounds were purified by semipreparative reversed-phase high-performance liquid chromatography (RP HPLC) (purity ≥ 95%) and their identity was confirmed by high-resolution mass spectrometry (ESI-HRMS).

The metabolic stability of analogs was assessed in the presence of the rat brain homogenate. All new cyclopeptides remained almost intact after 60 min incubation with the homogenate (Table 1).

The lipophilicity of compounds is often a useful characteristic allowing one to predict or rationalize their interactions with a given receptor [40]. Here, the liquid chromatography–mass spectrometry (LC-MS) procedure was applied to study the elution order of analogs in the reverse-phase chromatography to estimate their lipophilicity. The mixture of peptides **1**–**6** was run in the same experimental conditions on two types of columns, an Aeris Peptide C_18_ column and a novel biphenyl stationary phase HPLC column. The biphenyl column can be complementary to a C_18_ column, as it can engage in π–π interactions with the eluted compound [41]. As expected, peptides were more retained on the biphenyl column, due to the π stacking interactions of their aromatic rings with the solid support. However, the elution order of analogs **1**–**6** was identical (**5** < **1** < **3** < **4** = **6** < **2**) on both columns (Figure 1), demonstrating differences in their lipophilicity.

### 2.2. Biological Evaluation

Due to the strongly lipophilic character of the analogs, opioid receptor binding assays did not give reliable results. Therefore, the pharmacological profile of analogs **1**–**6** was evaluated in vitro in calcium mobilization functional assays at all three opioid receptors [42]. In this assay, changes in intracellular calcium level, monitored in Chinese hamster ovary (CHO) cells stably co-expressing opioid receptors and chimeric G proteins, reflect the activation of the GPCR and can be used for the pharmacological characterization of novel agonist and antagonist ligands [43,44].

The concentration–response curves of analogs **1**–**6** were obtained and the calculated agonist potencies (pEC_50_) and efficacies (α) of the analogs are summarized in Table 2. Dermorphin, DPDPE and dynorphin A were used as reference agonists for calculating the intrinsic activity at MOR, DOR and KOR, respectively. Out of the six novel analog peptides, **1**, **2**, **5** and **6** only marginally activated MOR, while **1** also activated KOR (EC_50_ values two and three rows of magnitude lower than the values obtained for the control MOR agonist dermorphin and KOR agonist dynorphin A, respectively).

Then, the analogs were tested as potential antagonists in inhibition response experiments at the MOR, DOR and KOR. Fixed concentrations of analogs were assayed against the concentration–response curves of the agonists, dermorphin, DPDPE and dynorphin A, respectively. Incubation of the cells stably expressing MOR or KOR with peptides **1**–**6** up to 1 µM concentration did not produce any effect on the concentration–response curve of the respective agonist (Figure 2A,C).

In the CHO cells stably expressing DOR, all analogs inhibited the maximal effect of DPDPE in a concentration-dependent manner and caused a slight shift in the concentration-response curves to the right (Figure 2B). To estimate the potency of the analogs as antagonists, pK_B_ values were calculated and compared with the value obtained for the selective DOR antagonist, naltrindole (Table 3). Naltrindole acted as a highly potent DOR antagonist with a pK_B_ value of 9.89. All tested compounds displayed a similar pharmacological profile. The strongest antagonist effect was observed for peptides **5**, **3**, **6** and **4** with pK_B_ values of 9.28, 9.17, 8.96 and 8.61, respectively. Weak correlation could be observed between the lipophilic character and the pK_b_ values of the analogs (with the exception of analog **1**), suggesting that although lipophilicity may be important, it is not the sole factor of binding ability and functional properties. Compound **5** was found to be the least lipophilic, as well as the strongest DOR antagonist. This is in agreement with the fact that polar and charged amino acid side chains are abundant in the DOR binding pocket and that strong polar interactions were observed between the bound ligands and the DOR previously [33,34].

The analogs **1**–**6** were designed as hybrids of two fragments, Dmt-Tic known to exert antagonism at DOR and cyclic fragments with high MOR affinity and were expected to possess MOR agonist/DOR antagonist properties. However, interference between the two epitopes may lead to the partial or even complete loss of affinity at the respective receptors. In this case, new analogs lost, almost completely, the ability to activate MOR but maintained their strong and selective DOR antagonist property.

### 2.3. Docking Studies

Molecular docking studies were performed in order to further assess the capacity of binding of synthetic peptide derivatives to the DOR and to reveal possible structure–activity relationships. The results of docking of the ligands indicated their potential to bind to DOR in both the active and inactive states with exceptionally high affinities. The predicted inhibitory constants, calculated from the binding free energies of docked complexes (Table 4) fell in the subnanomolar and low nanomolar range, indicating high pharmacological potential. The ligands were found to localize in the same binding pocket as the peptide agonist KGCHM07 [34] and peptide antagonist DIPP-NH_2_, [33] reported previously, and the orientations of ligands relative to the binding pocket were similar, regardless of the Dmt-Tic peptide bond configurations. However, markedly different binding modes were observed for compounds bound to the inactive and active DOR. In the former, the Tic side chain was engaged in a close hydrophobic contact with the Tyr109^2.64^ and/or Leu125^3.29^ side chains of the receptor (Figure 3A–C), whereas, in the active receptor, the Tic side chain was projected to the opposite direction, in the proximity of Ile277^6.51^, His278^6.52^ and Val281^6.55^ (Figure 3D–F). No significant or systematic differences were found between the *cis* and *trans* conformers and no clear trends could be identified between the sequence and structure of ligands with regard to the binding orientations and affinities. Close inspection of the docked complexes suggested that the pharmacologically relevant interaction between the ligands and the receptors is dominated by the Dmt-Tic epitope. In some complexes, the macrocyclic part of the molecules was observed to form secondary contacts at the entrance of the binding pockets, potentially contributing to the stability of the complex, resulting in low binding free energies. Conversely, in other receptor–ligand complexes, the macrocyclic part was found to be positioned partially outside of the binding pocket. This may provide explanation for the relative diversity of the predicted inhibitory constants.

Although the DOR antagonist properties of compounds **1**–**6**, as indicated by calcium mobilization data, cannot be directly inferred from in silico binding results, the different and specific binding modes observed for each structural and functional state of the DOR suggested further analysis.

Docked complexes of compounds **1**–**6** were compared to the x-ray crystallographic structure of the active DOR complexed with the peptide agonist KGCHM07 [34]. Poor alignment was observed between the pharmacophoric elements of DOR-bound KGCHM07 and the docked synthetic ligands (Figure 3B,C). The first major difference was that compounds **1**–**6** did not insert into orthosteric binding pocket as deep as KGCHM07, although such deeper penetration was previously proposed as an indication of receptor activation and the agonist property of the bound ligand (see reference above). The other conspicuous difference was that the Tic side chain of docked compounds was found to occupy the same space as the Dmt side chain of KGCHM07. Consequently, the Dmt side chain of compounds **1**–**6** was projected towards Asp210^5.35^ interfering with the salt bridge between that residue and Lys214^5.39^ and the electrostatic balance of the network of polar species in the binding pocket. No further pharmacophoric contacts identical or analogous to those described for KGCHM07 were observed, apart from the ionic interaction between Asp128^3.32^ and the *N*-terminal free amine of compounds **1**–**6**. Therefore, docked complexes of compounds **1**–**6** and the active state DOR could be deemed as false positive hits.

Similar to the active DOR-ligand complexes, docked complexes of the inactive DOR and compounds **1**–**6** were compared to the crystallographic structure of peptide antagonist DIPP-NH_2_ and the inactive state DOR. It is important to point out that the synthetic peptides examined in this study possess the same Dmt-Tic pharmacophore motif as DIPP-NH_2_. Interestingly, the orientation of Dmt-Tic was not found to be identical in compounds **1**–**6** to that of DIPP-NH_2_ (Figure 3E,F). While the Dmt side chains are relatively well aligned, the Tic side chain is positioned to the opposite direction from that of Tic of DIPP-NH_2_, to be in close contact with the Tyr109^2.64^ and/or Leu125^3.29^ side chains. This difference may be explained by the fact that the macrocyclic part of peptides **1**–**6** is relatively large and rigid compared to the structure of DIPP-NH_2_ and does not allow for facile and flexible insertion. Even so, relatively good alignment of aromatic side chains was found between the crystallographic structure of DIPP-NH_2_ and the docked ligands (Figure 3F). The Tic side chain of synthetic peptides **1**–**6** was found to occupy the same space as the Phe^3^ side chain of DIPP-NH_2_, whereas the Phe^4^ of the docked ligands was overlapping with the Tic side chain of DIPP-NH_2_. Based on this observation, it could be concluded that, in spite of the different binding mode of synthetic ligands, their antagonist property is provided through the analogous orientation of key pharmacophoric elements. The difference between binding modes does not exceed those previously observed between DIPP-NH_2_ and the morphinan antagonist naltrindole [33].

## 3. Experimental

### 3.1. General

Protected amino acids were purchased from Bachem AG (Bubendorf, Switzerland). *p*-Methylbenzhydrylamine (MBHA) resin was obtained from Novabiochem (San Diego, CA, USA). Semi-preparative and analytical RP-HPLC was performed using Waters Breeze instrument (Milford, MA, USA) with the dual absorbance detector (Waters 2487) on a Vydac C_18_ column (22 × 250 mm, 10 μm) and Vydac C_18_ column (4.6 × 250 mm, 5 μm), respectively. The LC-MS analysis was performed using Phenomenex Kinetex columns Aeris Peptide C_18_ and Biphenyl in a linear gradient from 5% to 80% B in A in 10 min (A: 0.1% (*v*/*v*) HCOOH in water and B: 0.1% (*v*/*v*) HCOOH in acetonitrile) using UHPLC Nexera coupled to IT-TOF Shimadzu instrument.

### 3.2. Peptide Synthesis

The linear precursor peptides of analogs **1**–**6** were prepared by the manual solid-phase technique using 9-fluorenylmethoxycarbonyl (Fmoc) protection for the α-amino groups, as described in detail elsewhere [45]. Peptides were assembled on MBHA resin (100–200 mesh, 0.8 mM/g,) to obtain C-terminal amides. Side-chain amino groups of d-Lys and d-Dap were protected by 4-methyltrityl (Mtt), β-carboxy group of Asp by 2-phenyl-isopropyl ester (*O*-2 Ph*i*Pr) and hydroxy group of Dmt by *t*-butyl (*t*-Bu). Removal of the Fmoc-protecting groups in each step was performed with 20% (*v*/*v*) piperidine in dimethylformamide (DMF) for 20 min. Coupling was achieved using 2-(1H-benzotriazol-1-yl)-1,1,3,3-tetramethyluronium tetrafluoroborate (TBTU) or 4-(4,6-dimethoxy-1,3,5-triazin-2-yl)-4-methylmorpholinium toluene-4-sulfonate (DMT/NMM/TSO^−^) and diisopropylethylamine (DIEA) as a neutralizing base. Fully assembled Fmoc-protected peptides were treated with 1% *v*/*v* trifluoroacetic acid (TFA) in dichloromethane (DCM) to remove side-chain groups (Mtt and *O*-2-Ph*i*Pr) from d-Lys/d-Dap and Asp, respectively, followed by on-resin cyclization (TBTU and DIEA). Following the removal of Fmoc group from Dmt, the cleavage of cyclized peptides from the resin was accomplished by the treatment with TFA/triisopropylsilane/water (95:2.5:2.5, *v*/*v*) for 3 h at room temperature. The crude peptides were purified by RP-HPLC on a Vydac C_18_ column (22 mm × 250 mm, 10 µm) using a linear gradient of 0–100% B over 15 min at the flow rate of 2 mL/min. Solvents: (A) 0.1% TFA in water and (B) 0.1% TFA in acetonitrile/water (80:20, *v*/*v*). The purity of the final peptides was verified by analytical RP-HPLC on a Vydac C_18_ column (4.6 × 250 mm, 5 µm,) in the same solvent system over 50 min. with the flow rate 1 mL/min. The synthesized compounds were characterized by ESI-MS (Appendix A).

### 3.3. Enzymatic Stability

Enzymatic degradation studies of the analogs were carried out using rat brain homogenate (10 mg protein/mL). To prepare, the homogenate rat brains were isolated, pooled and homogenized in a Polytron with 20 volumes of Tris.HCl (50 mM, pH 7.4), and the homogenate was then stored at −80°C until use. Aliquots of the homogenate (100 μL) were incubated with 100 μL of peptide (0.5 mM) over 0, 7.5, 15, 22.5, 30, and 60 min at 37 °C, in a final volume of 200 μL. The reaction was stopped at the required time by placing the tube on ice and acidifying the content with 20 μL of HCl (1 M). The aliquots were centrifuged at 20,000× *g* for 10 min at 4 °C. The supernatants were filtered over Millex-GV syringe filters (Millipore, Billerica, MA, USA) and analyzed by HPLC on a Vydac C_18_ column (4.6 mm × 250 mm, 5 μm,), using the solvent system of 0.1% TFA in water (A) and 80% acetonitrile in water containing 0.1% TFA (B) and a linear gradient of 0–100% B over 25 min. Three independent experiments for each assay were carried out. The amount of remaining peptide (area %) was calculated.

### 3.4. Cell Culture

All transfected cell lines (the generous gift from Prof. Girolamo Calo, Ferrara University, It) were maintained in culture medium consisting of Dulbecco’s MEM/HAM’S F-12 (50/50) supplemented with 10% fetal bovine serum (FBS) and streptomycin (100 μg/mL), penicillin (100 IU/mL), l-glutamine (2 mmol/L), geneticin (G418; 200 μg/mL), fungizone (1 μg/mL), and hygromycin B (100 μg/mL). Cells were kept at 37 °C in 5% CO_2_ humidified air. When confluence was reached (3–4 days), cells were sub-cultured as required using trypsin/EDTA and used for testing.

### 3.5. Calcium Mobilization Assay

Calcium mobilization assay was performed as reported previously [46]. CHO cells stably co-expressing human MOR or KOR and the C-terminally modified Gαqi5 and CHO cells co-expressing DOR and the GαqG66Di5 protein were used for the tests. Briefly, cells incubated for 24 h in 96-well black, clear-bottom plates were loaded with medium supplemented with probenecid (2.5 mmol/L), calcium-sensitive fluorescent dye Fluo-4 AM (3 μmol/L) and pluronic acid (0.01%) and kept at 37 °C for 30 min. Following aspiration of the loading solution and a washing step, serial dilutions of peptide stock solutions were added. Cells without peptides were treated as the control. Fluorescence changes were measured using the FlexStation II (Molecular Device, Union City, CA, USA. Maximal change in fluorescence, expressed as percent over the baseline fluorescence, was used to determine agonist response. Agonist potencies were given as pEC_50_ representing a negative logarithm of the molar concentration of an agonist that produces 50% of the maximal possible effect of that agonist. Concentration-response curves were fitted with the four parameter logistic nonlinear regression model:(1)$$Effect=\mathrm{baseline}+\frac{E_{max}-\mathrm{baseline}}{1+10^{\left(\log EC_{50}-X\right)\cdot n}}$$
where *X* is the agonist concentration and *n* is the Hill coefficient. Ligand efficacy was expressed as intrinsic activity (α) calculated as a ratio of the peptide Emax to Emax of the standard agonist.

In the antagonism-type experiments, peptides were injected into the wells 24 min. before adding an agonist. Then, standard antagonists, β-funaltrexamine, naltrindole and norbinaltorphimine for MOR, DOR and KOR, respectively, and peptides **1**–**6** were assayed at a single concentration against the concentration–response curve to the agonist. The antagonist potency was calculated from a double-reciprocal plot of equi-effective concentrations of agonist in the absence and presence of antagonist [47], and pK_B_ was derived from the equation: pK_B_ = log((slope − 1)/[A]).

Data were analyzed using one-way analysis of variance (ANOVA) followed by Dunnett’s post hoc test. Differences were considered statistically significant when *p* < 0.05. Data are expressed as mean ± SEM of *n* experiments.

### 3.6. Molecular Docking

In silico representations of the synthetic ligands were built manually using the Pymol and/or Avogadro software. The peptide bond connecting the Dmt and Tic residues has a considerable propensity to exist in both *cis*- and *trans*-configurations; therefore, ligands were built and dockings were performed accordingly.

The 2.8 Å resolution crystallographic structure of the DOR in complex with the peptide agonist KGCHM07 (pdb code: 6PT2) [34], and the 2.7 Å resolution crystallographic structure of the DOR bound to the bifunctional DOR antagonist and MOR agonist tetrapeptide DIPP-NH_2_ (4RWD) [33] were used as docking targets after missing sidechains were added.

Dockings were performed using the Autodock v.4.2 program, applying the Lamarckian Genetic Algorithm conformational search utility. All Φ, ψ, and χ^1^ ligand torsions, as well as binding pocket sidechains, in contact with the bound ligand, observed in the high-resolution structures, were kept flexible. Blind dockings of the peptide ligands were performed in an 80 Å × 80 Å × 80 Å grid volume, large enough to accommodate the whole receptor surface region accessible from the extracellular side. The spacing of grid points was set at 0.375 Å. A total of 1000 dockings were performed for each system. The clustering of docked poses was performed in order to assess the exhaustiveness of the conformational search. If the number of identified clusters exceeded 900, then the number of dockings was extended to 2000. After clustering, the resultant ligand-receptor complexes were ranked according to the corresponding binding free energies. The lowest energy bound states, in which specific conserved ligand–receptor interactions were observed were selected as potential binding modes. The above specified crystallographic structures of DOR were used as a reference. Binding free energies were converted into in silico inhibitory constants according to the following equation:

ΔG = RT ln K_i_(2)

## 4. Conclusions

The goal of this research was to obtain a series of analogs with an MOR agonist/DOR antagonist profile. It is known from the literature (as discussed in the Introduction) that numerous ligands with Dmt-Tic pharmacophore can stimulate MOR in conjunction with antagonism at DOR. The structures designed here contained the Dmt-Tic fragment attached to the cyclic scaffolds, which possessed high MOR affinity. However, the obtained ligands did not significantly activate any of the three opioid receptors but turned out to be potent and selective DOR antagonists.

It is well recognized that joining two molecules with different receptor affinities may cause in some cases interference between them, which may lead to the partial or even complete loss of affinity at the respective receptors. Analogs **1**–**6** lost almost completely their activity for the MOR but maintained their selective DOR antagonist property furnished by the Dmt-Tic epitope. The loss of MOR activity could be explained by the increased molecular size of compounds **1**–**6** which does not allow for facile insertion into the MOR binding pocket. Conversely, analogous binding orientations of the Dmt-Tic epitope observed for compounds **1**–**6**, as compared to the crystallographic structure of DOR-bound DIPP-NH_2_, provides explanation for the remarkable DOR antagonist potencies. Analog **5**, being the strongest antagonist of the series, could be further developed as a pharmacological tool for investigating the role of DOR in pain control.

## Figures and Tables

**Figure 1 molecules-25-04260-f001:**
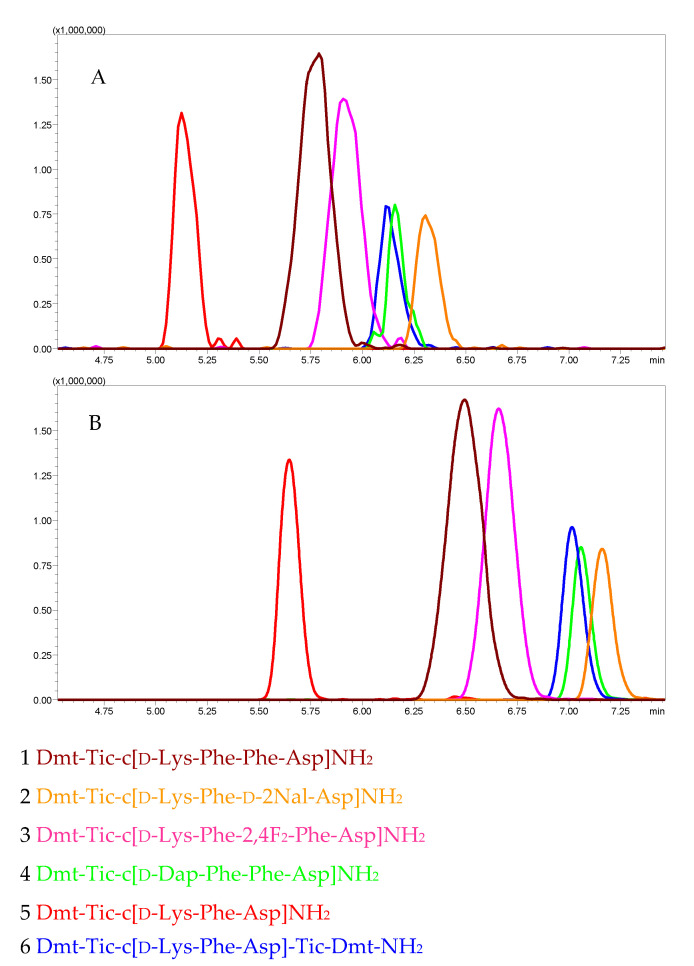
LC-MS chromatograms (extracted ion currents (XIC)) for peptides **1**–**6**. Panel (**A**): Aeris Peptide XB-C_18_ column. Panel (**B**): Kinetex Biphenyl column. Gradient: 5–80% acetonitrile in water in 10 min., both solvents contained 0.1% HCOOH.

**Figure 2 molecules-25-04260-f002:**
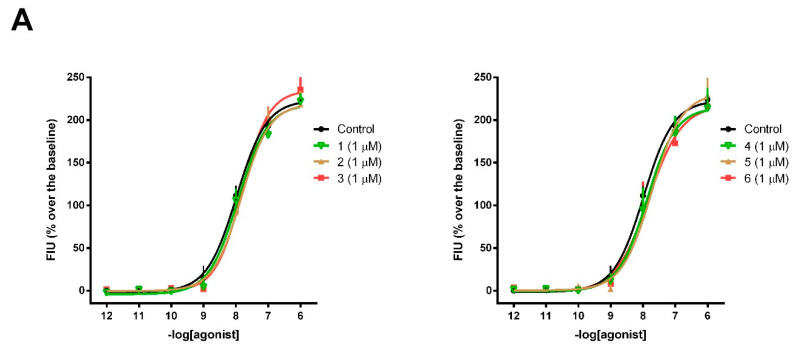
Calcium mobilization assay. Concentration–response curves to dermorphin (**A**), DPDPE (**B**) and dynorphin A (**C**) obtained in the absence (control) and presence of the tested compounds. Data are the mean ± SEM of 3 separate experiments made in duplicate, *p* < 0.05 vs. control, according to one-way analysis of variance (ANOVA) followed by the Dunnett’s post hoc test. FIU, fluorescence intensity units.

**Figure 3 molecules-25-04260-f003:**
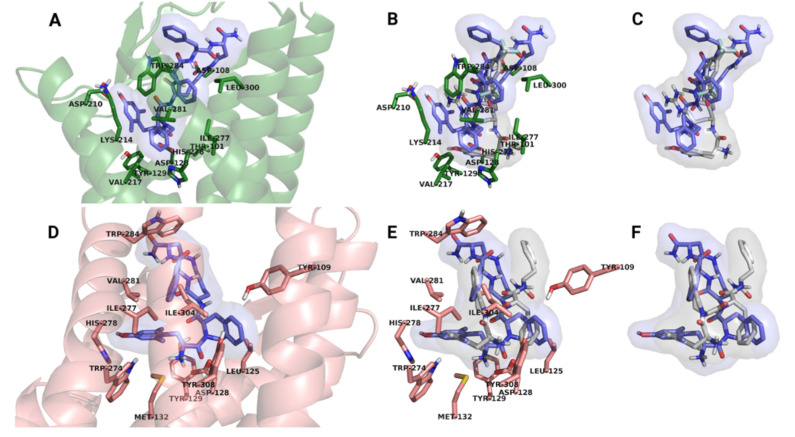
Representative low-energy docking poses of the examined compounds with active (**A**–**C**) and inactive DOR (**D**–**F**). Compounds **1** for the active and **5** for the inactive DOR are shown as examples. Amino acid side chains that line the binding pocket of active and inactive DOR are shown in pink and green stick representations, respectively. Non-polar hydrogen atoms are omitted for clarity.

**Table 1 molecules-25-04260-t001:** Structure, monomer/dimer ratios, yield and enzymatic stability of analogs **1**–**6**.

No.	Sequence	Ring Size	Monomer/ Dimer Ratio	Yield [%]	Enzymatic Stability Area [%] ^a^
**1**	Dmt-Tic-c[d-Lys-Phe-Phe-Asp]NH_2_	17	2.9	28	97.88 ± 0.83
**2**	Dmt-Tic-c[d-Lys-Phe-d-2Nal-Asp]NH_2_	17	2.8	25	98.17 ± 0.23
**3**	Dmt-Tic-c[d-Lys-Phe-2,4F_2_-Phe-Asp]NH_2_	17	3.1	25	95.19 ± 0.86
**4**	Dmt-Tic-c[d-Dap-Phe-Phe-Asp]NH_2_	14	1.2	18	96.21 ± 0.89
**5**	Dmt-Tic-c[d-Lys-Phe-Asp]NH_2_	14	0.4	11	96.76 ± 0.91
**6**	Dmt-Tic-c[d-Lys-Phe-Asp]-Tic-Dmt-NH_2_	14	1.6	19	97.61 ± 1.20

^a^ Amount of peptide remained after 60 min incubation with rat brain homogenate.

**Table 2 molecules-25-04260-t002:** Agonist potencies (pEC_50_) and efficacies (α) of analogs **1**–**6** determined on the µ-opioid receptor (MOR), δ-opioid receptor (DOR) and κ-opioid receptor (KOR) coupled with calcium signaling via chimeric G proteins.

Peptide	MOR		DOR		KOR	
	pEC_50_ (CL_95%_)	α ± SEM	pEC_50_ (CL_95%_)	α ± SEM	pEC_50_ (CL_95%_)	α ± SEM
**dermorphin**	8.66 ± 0.10	1.00	inactive		inactive	
**DPDPE**	inactive		7.32 ± 0.18	1.00	inactive	
**dynorphin A**	6.67 ± 0.50	0.83 ± 0.10	7.73 ± 0.27	0.99 ± 0.04	9.04 ± 0.09	1.00
**1**	6.18 ± 0.51	0.15 ± 0.02	inactive		6.31 ± 0.59	0.20 ± 0.06
**2**	6.21 ± 0.5	0.37 ± 0.05	inactive		inactive	
**3**	inactive		inactive		inactive	
**4**	inactive		inactive		inactive	
**5**	6.09 ± 0.17	0.3 ± 0.03	inactive		inactive	
**6**	6.48 ± 0.49	0.17 ± 0.20	inactive		inactive	

Inactive means that the compound was inactive up to 10 μM. All values are expressed as mean ± SEM, *n* ≥ 5.

**Table 3 molecules-25-04260-t003:** Antagonist potencies (pK_B_) of analogs **1**–**6** and naltrindole.

No	Sequence	pK_B_(CL_95%_)
**1**	Dmt-Tic-c[d-Lys-Phe-Phe-Asp]NH_2_	7.37 ± 0.29
**2**	Dmt-Tic-c[d-Lys-Phe-d-2Nal-Asp]NH_2_	7.55 ± 0.32
**3**	Dmt-Tic-c[d-Lys-Phe-2,4F_2_-Phe-Asp]NH_2_	9.17 ± 0.35
**4**	Dmt-Tic-c[d-Dap-Phe-Phe-Asp]NH_2_	8.61 ± 0.15
**5**	Dmt-Tic-c[d-Lys-Phe-Asp]NH_2_	9.28 ± 0.34
**6**	Dmt-Tic-c[d-Lys-Phe-Asp]Tic-Dmt-NH_2_	8.96 ± 0.28
**naltrindole**		9.89 ± 0.12

**Table 4 molecules-25-04260-t004:** Predicted binding free energies and inhibitory constants of bivalent opioid ligands with the active and inactive DOR.

No	Sequence	Predicted Receptor Affinity (K_i_/pM)			
		DOR			
		Active State		Inactive State	
		*cis*	*trans*	*cis*	*trans*
**1**	Dmt-Tic-c[d-Lys-Phe-Phe-Asp]NH_2_	3.9	159.0	6500	211.5
**2**	Dmt-Tic-c[d-Lys-Phe-d-2Nal-Asp]NH_2_	681.9	7170	9660	380.0
**3**	Dmt-Tic-c[d-Lys-Phe-2,4F_2_-Phe-Asp]NH_2_	7920	4930	4010	1150
**4**	Dmt-Tic-c[d-Dap-Phe-Phe-Asp]NH_2_	2640	1190	9040	6030
**5**	Dmt-Tic-c[d-Lys-Phe-Asp]NH_2_	1870	5700	3200	9640
**6**	Dmt-Tic-c[d-Lys-Phe-Asp]Tic-Dmt-NH_2_	1460	3310	520.5	540.8

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
