# Peer review of "Design, Synthesis and Functional Analysis of Cyclic Opioid Peptides with Dmt-Tic Pharmacophore"

_molecules, 2020, doi:10.3390/molecules25184260_

Round 1

Reviewer 1 Report

The manuscript presents some informations on novel hybrids with dual activity towards MOR and DOR opioid receptors.

Overall, this paper is well-written and very intersting. However, in my opinion one crucial data is missing. In fact, the Authors wrote "pharmacological assesment" (the title of the paper), but cacium immobilization method is quite weak to serve (only) as a technique which gives any info about phramacological profile of the drug. In line to this, can the Authors provide with any results (e.g. in vivo, especially considered that every drug is composed of elements related to the opioid system, thus pain, etc.) that would strongly indicate of the "pharmacological assesment"?

Otherwise, the title should be modified

in my opinion, also studies on the conformational and/or enzymatic stability would strongly improve this paper, as still these drugs are peptides.

Author Response

We would like to express our gratitude for the critical remarks which helped us improve the manuscript.

  1. The manuscript presents some informations on novel hybrids with dual activity towards MOR and DOR opioid receptors. Overall, this paper is well-written and very intersting. However, in my opinion one crucial data is missing. In fact, the Authors wrote "pharmacological assesment" (the title of the paper), but cacium immobilization method is quite weak to serve (only) as a technique which gives any info about phramacological profile of the drug. In line to this, can the Authors provide with any results (e.g. in vivo, especially considered that every drug is composed of elements related to the opioid system, thus pain, etc.) that would strongly indicate of the "pharmacological assesment"? Otherwise, the title should be modified

 Answer:

The compounds were not soluble enough to perform in vivo tests. So, according to the     Reviewer’s suggestion, the title has been changed, pharmacological assessment has been  replaced by functional analysis.

  1. In my opinion, also studies on the conformational and/or enzymatic stability would strongly

     improve this paper, as still these drugs are peptides.

Answer:

As suggested, enzymatic stability has been studied using rat brain homogenate as a source of proteolytic enzymes. As could be expected on the bases of our earlier experience with cyclic peptides, the analogs were very stable and remained almost completely intact after 60 min incubation. For that reason the results are given as the area % and not half-life (Table 1).

As to the conformational stability, these cyclic peptides allow for very little variation of backbone conformation. According to our earlier studies, where detailed conformational analysis was performed on a set of analogous peptides (Adamska-BartÅ‚omiejczyk et. al.Bioorg Med Chem Lett. 2017 Apr 15;27(8):1644-1648. doi: 10.1016/j.bmcl.2017.03.016.) we have found, that  flexibility of the macrocycle has negligible impact on the ability of these
peptides to bind to the receptors. It is more likely to be governed by conformational flexibility and three-dimensional arrangement of the pharmacophore groups (protonated N-terminal amine, aromatic side chains) Such arrangements, namely the different combinations of spatial positions of pharmacophores were analyzed exhaustively in the present study. This was achieved through having all pharmacophore side chains as well as binding pocket residues treated fully flexible during dockings.

Reviewer 2 Report

Manuscript ID: molecules-925980

Molecules

Title: Design, synthesis and pharmacological assessment of cyclic opioid peptides with Dmt-Tic pharmacophore

General comments:

The authors report the synthesis and pharmacological evaluation of six new hybrids of the Dmt-Tic pharmacophore combining cyclic scaffolds which were previously shown to possess δ opioid receptor DOR-antagonist properties and high MOR affinity, respectively. The purpose of the research was obtain a series of analogs with MOR agonist/DOR antagonist profile; however, all the obtained ligands turned out to be selective antagonist of the δ opioid receptor (DOR) and did not activate or block µ opioid receptor (MOR) or κ opioid receptor (KOR).

The loss of MOR activity could be explained by the increased molecular size of compounds 1-6 which does not allow for facile insertion into the MOR binding pocket. Moreover, docking studies indicated that these compounds attach to the DOR in a slightly different orientation, suggesting further analysis.

In conclusion, the authors assess that compound 5, being the best antagonist of the series, could be further developed as a pharmacological tool for investigating the role of DOR in pain control.

The overall approach of the work is good, the topic is original and could be interesting. The article’s technical quality is fair. The language is clear and the reading easily understandable. The Abstract is concise and clear. The methods are rather well described, although some interpretations and the conclusions are not perfectly clear. Indeed, you declare

Additional revisions:

  • Page 4, line line 143-145: “Incubation of the cells stably expressing MOR or KOR with peptides 1-6 up to 1 μM concentration did not produce any effect on the concentration-response curve of the respective agonist.” It should be inserted, in the manuscript or in the supporting information, the figure with the concentration-response curves of the hybrids compounds also for MOR and KOR.
  • Page 5, line 154-155: It could be considered also compound 4 a good DOR-antagonist with a pKB value of 8.61, as shown in table 3 at page 7.
  • Page 7, line 203 and page 8 line 218-224: Figure 4 was reported in the text but it is not present in the manuscript or in the supporting information. Please, provide it.
  • Page 7, line 171-173: “In this case, new analogs lost, almost completely, the ability to activate MOR but maintained their strong and selective DOR antagonist property.” You declare that the new compounds lost the ability to activate MOR; in contrast, at page 4 line 139-140 you affirm that “ Out of the six novel analogs peptide 1 very weakly activated MOR and KOR, while 2, 5 and 6 only MOR”. Please, explain it.
  • Page 9, line 247-248: “Analogs 1-6 lost almost completely their affinity to the MOR but maintained their selective DOR antagonist property furnished by the Dmt-Tic epitope….”. You speak about affinity to the MOR but it is not reported the opioid receptor binding assay for the new compounds. It could be appropriate replace affinity with MOR “activity” or “property”.
  • Page 14, Figure 2: It is recommended to include a negative control in the experimental design (wash buffer) that did not elicit a fluorescent signal, in order to indicates that the medium in which the compounds are dissolved was free of any contaminants that could influence the results.
  • Page S4, table S1: In the caption you specify what “b” refers to but it is missing in the table.

Author Response

We would like to express our gratitude for their critical remarks which helped us improve the manuscript.

General comments:

The authors report the synthesis and pharmacological evaluation of six new hybrids of the Dmt-Tic pharmacophore combining cyclic scaffolds which were previously shown to possess δ opioid receptor DOR-antagonist properties and high MOR affinity, respectively. The purpose of the research was obtain a series of analogs with MOR agonist/DOR antagonist profile; however, all the obtained ligands turned out to be selective antagonist of the δ opioid receptor (DOR) and did not activate or block µ opioid receptor (MOR) or κ opioid receptor (KOR).

The loss of MOR activity could be explained by the increased molecular size of compounds 1-6 which does not allow for facile insertion into the MOR binding pocket. Moreover, docking studies indicated that these compounds attach to the DOR in a slightly different orientation, suggesting further analysis.

In conclusion, the authors assess that compound 5, being the best antagonist of the series, could be further developed as a pharmacological tool for investigating the role of DOR in pain control.

The overall approach of the work is good, the topic is original and could be interesting. The article’s technical quality is fair. The language is clear and the reading easily understandable. The Abstract is concise and clear. The methods are rather well described, although some interpretations and the conclusions are not perfectly clear.

Additional revisions:

  1. Page 4, line line 143-145: “Incubation of the cells stably expressing MOR or KOR with peptides 1-6 up to 1 μM concentration did not produce any effect on the concentration-response curve of the respective agonist.” It should be inserted, in the manuscript or in the supporting information, the figure with the concentration-response curves of the hybrids compounds also for MOR and KOR.

Answer:

In Figure 2 two additional graphs have been added, showing concentration response curves to dermorphin (A) and dynorphin A (C).

  1. Page 5, line 154-155: It could be considered also compound 4 a good DOR-antagonist with a pKB value of 8.61, as shown in table 3 at page 7.

Answer:

Compound 4 has been added.

  1. Page 7, line 203 and page 8 line 218-224: Figure 4 was reported in the text but it is not present in the manuscript or in the supporting information. Please, provide it.

Answer:

The figure number was incorrect. Should be Figure 3 not Figure 4 in these places. It has been corrected in the revised manuscript.

  1. Page 7, line 171-173: “In this case, new analogs lost, almost completely, the ability to activate MOR but maintained their strong and selective DOR antagonist property.” You declare that the new compounds lost the ability to activate MOR; in contrast, at page 4 line 139-140 you affirm that “ Out of the six novel analogs peptide 1 very weakly activated MOR and KOR, while 2, 5 and 6 only MOR”. Please, explain it.

Answer:

The sentence has been changed and is now written more clearly; "Out of the six novel analogs peptides 1, 2, 5 and 6 only marginally activated MOR, while 1 also KOR (EC50 values two and three rows of magnitude lower than the values obtained for control MOR agonist dermorphin and KOR agonist dynorphin A, respectively)."

  1. Page 9, line 247-248: “Analogs 1-6 lost almost completely their affinity to the MOR but maintained their selective DOR antagonist property furnished by the Dmt-Tic epitope….”. You speak about affinity to the MOR but it is not reported the opioid receptor binding assay for the new compounds. It could be appropriate replace affinity with MOR “activity” or “property”.

Answer:

Indeed, that was a mistake. Affinity has been replaced by activity.

  1. Page 14, Figure 2: It is recommended to include a negative control in the experimental design (wash buffer) that did not elicit a fluorescent signal, in order to indicates that the medium in which the compounds are dissolved was free of any contaminants that could influence the results.

Answer:

There was always a control in each experiment, containing only buffer (no peptide added). A sentence has been added in the Methods section: "Cells without peptides were treated as control".

  1. Page S4, table S1: In the caption you specify what “b” refers to but it is missing in the table.

Answer:

The caption has been added.

Round 2

Reviewer 2 Report

The authors responded comprehensively to all requests and comments.